# Incretin Mimetics as Potential Therapeutics for Concussion and Traumatic Brain Injury: A Narrative Review

**DOI:** 10.3390/ijms27010045

**Published:** 2025-12-20

**Authors:** Samuel Sipos, Mirjana Jerkic, Ori D. Rotstein, Tom A. Schweizer

**Affiliations:** 1The Keenan Research Centre for Biomedical Science of St. Michael’s Hospital, Unity Health Toronto, 30 Bond Street, Toronto, ON M5B 1W8, Canada; samuelsipos23@rcsi.ie (S.S.);; 2Royal College of Surgeons in Ireland, 123 St Stephen’s Green, D02 YN77 Dublin, Ireland; 3The Division of General Surgery, St. Michael’s Hospital, Department of Surgery University of Toronto, 149 College Street, Toronto, ON M5T 1P5, Canada; 4The Division of Neurosurgery, St. Michael’s Hospital, Department of Surgery, University of Toronto, 149 College Street, Toronto, ON M5T 1P5, Canada

**Keywords:** traumatic brain injury (TBI), GLP-1 (glucagon-like peptide-1), GIP (gastric inhibitory peptide or glucose-dependent insulinotropic peptide), Gcg (glucagon), neuroinflammation, concussion, incretin mimetic

## Abstract

Traumatic brain injury (TBI) represents a significant health concern, with an estimated 70 million annual cases worldwide. Mild brain trauma (concussions) is the most common TBI (81%), followed by moderate (11%) and severe (8%). Cytokine release and neuroinflammation after TBI may cause blood–brain barrier and tissue damage, triggering unfavorable outcomes, including disabilities and mortality. Current TBI treatments, focused on preventing secondary injury, are limited and insufficient. Therefore, new therapeutic approaches are necessary. A growing body of recent literature supports the potential use of incretins: glucagon-like peptide-1, glucose-dependent insulinotropic peptide, and glucagon receptor agonists, as potent neurotrophic/neuroprotective agents. Experiments performed in cellular and animal models, and a limited number of clinical studies, provide evidence that incretins might be a novel and effective treatment for TBI. Incretin-based compounds have already been shown to be safe and efficacious for the treatment of type 2 diabetes mellitus in humans. Therefore, incretins are ideal candidates for rapid evaluation in clinical trials of TBI and might become a novel therapeutic tool for a condition that has very few disease modifying treatments available. Well-designed human clinical trials are urgently needed to determine optimal dosing, timing, and patient selection for effective incretin use in concussion and TBI.

## 1. Introduction

Traumatic brain injury (TBI) is a major global health problem, affecting an estimated 70 million people annually and is a leading cause of disability and mortality worldwide [1]. Mild TBI (mTBI), or concussion, accounts for approximately 80% of all cases [2,3] and is increasingly recognized for its potential to cause persistent neurological symptoms and long-term complications. Currently, treatment for TBI is mainly symptomatic and there is a lack of disease-modifying treatments, especially in mTBI [4]. As a result, many patients experience incomplete recovery, persistent neurological deficits, and long-term complications such as cognitive impairment and mood disorders [5]. This highlights an urgent need for therapies that directly target the biological mechanisms of secondary injury and promote brain repair, particularly for concussion, where treatment options remain extremely limited.

Glucagon-like peptide-1 receptor agonists (GLP-1RAs) and other incretin-based drugs are well-established treatments for type 2 diabetes mellitus (T2DM) and obesity due to their metabolic effects and favorable safety profiles [6]. Some benefits of GLP-1RAs are driven by weight loss, while other effects seem to be independent of weight or glucose control. These medications are showing promise as treatments for liver and kidney inflammatory conditions, cardiovascular disorders, osteoarthritis, and beyond [7]. Emerging evidence suggests that GLP-1RAs also have potent neuroprotective actions [8]. Recent studies, mainly in the form of pre-clinical trials, have shown that their neurotrophic and neuroprotective properties may be useful for treating/lessening the sequelae of TBI. It is proposed that incretins exert these effects through various mechanisms including reduction in apoptosis, oxidative stress, and neuroinflammation as well as improving mitochondrial function and neuronal survival [4].

We performed an overview of the current literature on the neuroprotective and neurotrophic properties of incretin mimetics in the management of TBI and neuroinflammation. This review consolidated results from pre-clinical trials and explored potential mechanistic pathways involved in protective effects. Additionally, this review elucidated the mechanistic and practical gaps in knowledge present among the reviewed literature and aims to recognize what further steps could be taken to help truly understand if incretin mimetics could be beneficial for clinical use in TBI and neuroinflammation.

### TBI and Incretin Mimetics

TBI pathology consists mainly of two types of injuries: primary and secondary injuries. Primary injuries are injuries that are a direct result of the initial trauma and often manifest as stretching, shearing, and mechanical deformation of blood vessels and neuronal tissue, which can result in hemorrhages, contusions, and hematomas. Injuries that occur in the minutes to days after the initial trauma are classified as secondary injuries and result from several inflammatory, molecular, and chemical processes initiated by primary injury [9].

Due to the instant nature of primary injuries, there is little room for using pharmacological agents to prevent or slow the progression of primary injuries. However, due to the longer manifestation time of secondary injuries ranging from minutes to days or years, there is a therapeutic window for pharmacological intervention to mitigate or prevent secondary injury [10].

Incretin mimetics are drugs that mimic the effects of incretin hormones. They are widely used to treat T2DM and obesity due to their ability to stimulate insulin release and regulate appetite. However, the recent literature suggests they have neuroprotective and neurotrophic properties that can aid in the treatment of many brain diseases and injuries including TBI and neuroinflammation [11].

In this context, it is important to examine the pharmacokinetics and blood–brain barrier (BBB) penetration of incretin mimetics, as these factors may influence their therapeutic effectiveness in TBI and other neurological conditions. However, existing studies on the pharmacokinetics and pharmacodynamics of incretins are largely limited to healthy volunteers [12] or populations with diabetes, obesity [13,14], or kidney impairment [15], and remain insufficient for TBI and other brain disorders.

Salameh et al. [16] explored the pharmacokinetics and brain uptake of intravenously (i.v.) given radio-labeled incretin receptor agonists (IRAs) in mice (exendin-4, liraglutide, lixisenatide, and semaglutide) during a 1 h period. They found that the non-acylated and non-PEGylated IRAs had better blood-to-brain influx than the acylated IRA molecules, mediated by more efficient transcytosis through brain endothelial cells. Bader et al. found that a sustained-release exenatide formulation was reaching the central nervous system (CNS) and improving visual and spatial deficits in a mTBI mouse model [17]. However, some IRAs, such as tirzepatide, do not cross the BBB immediately after intravenous administration. Instead, they enter at a slower rate, over approximately six hours, likely through extracellular pathways [18]. In addition, TBI and neuroinflammation can disrupt BBB integrity [19], which may increase drug penetration [20], including the penetration of incretin mimetics. Overall, further research is needed to better characterize the mechanisms of action and pharmacokinetics [21] of incretin agonists in individuals with TBI, as well as to determine the most effective delivery routes. Intranasal administration, in particular, represents a promising approach for improving drug access to brain targets [22].

## 2. Results—Incretin Mimetic Use in Concussion, TBI, and Neuroinflammation

The outcomes measured by all studies considered for this review are shown in Table 1. These studies encompass both in vivo pre-clinical animal models as well as in vitro studies aimed at elucidating possible mechanism of action. Some studies (4/21) included in the review modeled subarachnoid and intracerebral hemorrhage—both common sequelae—induced by TBI. Table 1 also details what incretin mimetic was used and its timeline of application. The majority of the studies (16/21) included an in vivo component only, while 5/21 reported both in vivo and in vitro experiments.

Studies listed in Table 1 used several methods to measure memory deficits. The novel object recognition task was used to measure visual memory [17,23,24,26,27,28,32,34,35,38]. The Morris water maze [28,30,31,32,41] or Y-maze [17,23,24,25,27,38] was frequently used to measure spatial memory. Studies that measured memory impairment using maze and recognition-based tests saw that animals challenged with TBI performed significantly worse than control animals on the memory tasks. This deficit in performance was often recovered to near control levels in the TBI + incretin mimetic animal groups across the studies (Table 1 and Table 2). In some of the moderate-severe TBI models, motor impairment was also measured by several tests within each model and across different models: composite neuroscore [29], circling test [29], paw placement test [29], modified neurological safety score [31], forelimb placement test [31], rotarod performance test [31], sticky tape test [34], error ladder walking test [34], and modified Garcia test [40,41,42]. Similarly to the memory tasks, incretin mimetics reduced motor deficits (Table 1 and Table 2).

In vitro models measured oxidative stress and excitotoxicity using neuronal cell viability. When neuroblastoma cells or rat primary neurons were challenged with excess hydrogen peroxide or glutamate, there was a decrease in neuronal cell viability below control levels. When these cells were treated with an incretin mimetic, there was a significant recovery in neuronal cell viability back to near control levels, therefore alleviating excitotoxic and oxidative insult (Table 1 and Table 2) [23,25,27,30]. In vivo models measured oxidative stress using several label-based optical assays (fluorescence and chemiluminescence) to determine the level of reactive oxygen species (ROS) species generation. When animals were challenged with TBI there was an increase in ROS generation above control levels. When animals were treated with an incretin mimetic there was a significant reduction in ROS generation back to near control levels, therefore alleviating oxidative stress (Table 1 and Table 2) [31,33,34,39,40].

Microglial activation and astrogliosis were assessed using immunohistochemistry and measured by ionized calcium-binding adaptor molecule 1 (Iba1) and glial fibrillary acidic protein (GFAP), respectively. In papers that measured inflammation, when animals were challenged with TBI, both markers were increased above control levels [17,24,28,32,33,34]. Increased microglial activation was significantly reduced back to near control levels in the TBI + incretin mimetic animal groups. For astrogliosis, some studies found that there were no significant reductions in astrogliosis when animals were treated with incretin mimetics [17,24,33,34], while others found significant reductions in astrogliosis [28,32] (Table 1 and Table 2). BBB permeability was measured both directly by using Evans Blue dye extravasation and indirectly through cerebral edema measurements. For both measurements, there were increases when animals were challenged with TBI. However, when animals were treated with incretin mimetics, there were significant decreases in Evans blue dye extravasation and cerebral edema (Table 1 and Table 2) [29,32,39,40,42].

## 3. Discussion

### 3.1. TBI Pathophysiology

TBI pathology differs somewhat depending on the severity of the injury, but the mechanisms underpinning secondary injury remain largely similar. TBI pathology, while varying with injury severity, involves mechanisms such as glutamate excitotoxicity, ionic imbalance, mitochondrial dysfunction, oxidative stress, BBB disruption, and neuroinflammation, all of which can contribute to cell death [43,44]. Importantly, these processes provide multiple points at which incretin mimetics may exert neuroprotective effects. In mTBI, the external forces imparted on the brain cause neuronal tissue to stretch but do not cause any macroscopic damage. In moderate–severe TBI, instead of tissue stretching, there is more shearing and deformation, causing macroscopic brain damage.

Following TBI, there is a release of excitatory neurotransmitters, namely glutamate. Normally, excess glutamate is buffered by astrocytes to prevent excitotoxicity, but due to several factors—mainly transporter dysfunction and changes in astrocyte morphology—excess glutamate is unable to be diminished by astrocytes. The excess glutamate stimulates receptors, causing membrane depolarization and contributing to uncontrolled ion influx, particularly calcium. This ion influx causes enzymatic activation, which damages the membrane, mitochondria, cytoskeleton, and DNA, ultimately causing cell death [45].

Inappropriate ion flux following TBI can cause ionic imbalances that initiate damaging processes. High levels of intracellular calcium ions are acutely accommodated by sequestering excess calcium ions in mitochondria. However, long-term, high levels of intracellular calcium ions can cause the generation of ROS, mitochondrial membrane impairment, and activation of pro-apoptotic enzymes. All occurrences lead to the inability of mitochondrial ATP synthesis, metabolic dysfunction, structural damage, and ultimately cell death [43,46].

#### Neuroinflammation

The initial trauma from TBI causes elevation of pro-inflammatory mediators and subsequent activation of neutrophils, resident microglia, and astrocytes, which drive the process of neuroinflammation. Astrocytes chiefly help preserve neuronal tissue, while microglia produce inflammatory signals and clear cellular debris. The inflammatory response is both healing and harmful for the brain. If regulated, the inflammatory response can help remove noxious debris and initiate cellular repair. However, if dysregulated, it can achieve the opposite by causing tissue damage, hindering cellular repair, and cell death. Neuroinflammation is also present in other injury mechanisms, causing potentially harmful positive feedback loops. Inflammation can dysregulate BBB permeability, which can further worsen injury. Inflammatory induced changes in astrocyte morphology and glutamate receptor expression can contribute to excitotoxicity. Microglia are also directly responsible for the generation of some ROS, contributing to oxidative stress [47,48].

Neuroinflammation is a prominent component in TBI due to its persistent involvement in injury mechanisms, but also its ability to cause long-term complications. TBI has been associated with the development of several harmful sequelae with inflammation being an important underlying factor. The development of dementia, Alzheimer’s disease, chronic traumatic encephalopathy (CTE), movement disorders, and psychiatric consequences are partly attributable to the activation of microglia and persistent neuroinflammation [49,50].

### 3.2. Common Outcomes of Incretin Use in TBI and Neuroinflammation

Across the reviewed studies, incretins exerted multiple beneficial effects on the adverse biological consequences of TBI. Most notably, treatment with incretin mimetics consistently improved TBI-related memory and sensorimotor deficits. In many cases, performance impairments on memory and motor tasks were restored to near control levels following incretin-based therapy (Table 1 and Table 2). However, cognitive assessments varied widely across studies [17,23,24,25,26,27,28,29,30,31,32,34,35,38,40,41,42], making cross-study comparisons challenging and highlighting the need for standardized outcome measures.

Deeper insight into the mechanism of incretin mimetic effects in TBI revealed that incretin mimetics were able to alleviate oxidative stress and excitotoxicity. The results from in vitro models demonstrated that the reduction in cell viability, due to simulated excitotoxicity and oxidative stress, was almost completely restored with treatment using an incretin mimetic (Table 1 and Table 2). Similarly, in vivo models generally showed a reduction in TBI-induced ROS generation and a marked decrease in oxidative stress with incretin mimetic treatment (Table 1 and Table 2). Because excessive oxidative stress contributes to neuronal apoptosis [51], mitochondrial dysfunction [52,53], neuroinflammation [53,54], and BBB disruption [52] after TBI, the ability of incretins to reduce ROS production may be especially important for improving TBI outcomes and prognosis.

In several studies, incretin mimetics were able to reduce the inflammatory response and ameliorate the increase in BBB permeability induced by TBI through their effect on microglia and possibly astrocytes. The results generally show that elevated microglial activation from TBI was reduced back to near control levels with incretin treatment (Table 1 and Table 2) indicating that incretin mimetics have the potential to attenuate the inflammatory response associated with TBI. However, findings regarding the ability of incretin mimetics to reverse TBI-induced astrogliosis are inconsistent. Some studies report significant improvement [28,32], whereas others show minimal or no recovery [17,24,33,34], highlighting the need for further investigation before firm conclusions can be drawn. In contrast, the evidence is more consistent for BBB integrity. Across studies, incretin mimetics reliably improved TBI-related increases in BBB permeability (Table 1 and Table 2). This effect is particularly important, as BBB disruption is a major contributor to secondary injury mechanisms following TBI [55].

### 3.3. Mechanisms Underlying Neuroprotective and Neurotrophic Properties of Incretin Mimetics

Studies reviewed offer several mechanisms by which incretin mimetics achieved their neuroprotective and neurotrophic effects. Several in vitro studies partially attributed their positive results to incretin mimetics’ ability to stimulate anti-apoptotic and pro-survival pathways [23,24,25,27,31,33,40,41]. It is proposed that incretin mimetics achieved these effects through stimulation of incretin receptors (GLP-1 and GIP receptors), which causes increased cAMP production and subsequent activation of the cAMP response element binding protein (CREB) pathway. Activation of the CREB pathway results in reductions in caspase-3 activation, ROS production, and Bax/Bcl-2 ratios. These factors lead to an inhibition of apoptosis (Figure 1). CREB activation also promotes cell survival through up-regulation of cyclin, inflammation, and insulin-related genes. Most studies attributed upstream CREB activation to the protein kinase A (PKA) pathway, but one attributed it to the extracellular signal-regulated kinase (ERK) pathway [31]. While CREB activation was mainly ascribed to the resultant anti-apoptotic and pro-survival effects, some studies found that other interconnected pathways were also contributors. Activation of the phosphoinositide 3-kinase (PI3K)/protein kinase B (Akt) pathway was proposed to produce some similar anti-apoptotic and pro-survival effects [24,41]. Similarly, mitogen-activated protein kinase (MAPK) pathways were proposed to produce similar effects as well [24,25,31,41]. However, there was a lack of clear identification of how these other pathways contributed. Ultimately, incretin mimetics’ anti-apoptotic and pro-survival effects were ascribed to decreased neuronal degeneration, excitotoxicity, oxidative stress, and memory deficit outcomes (Figure 1).

Several studies also partially attributed their beneficial outcomes to incretin mimetics’ ability to ameliorate neuroinflammation and BBB dysfunction [17,24,26,28,29,32,34,39,40,42]. It is proposed that incretin mimetics primarily achieved these effects through decreased microglial activation and astrogliosis. Several studies proposed these effects are also mediated through the CREB, PI3K/Akt, and MAPK pathways by incretin receptor stimulation. Additionally, some studies suggest anti-inflammatory effects were also generated through non-incretin receptor dependent pathways like activation of the AMP-activated protein kinase (AMPK) and subsequent inhibition of the pro-inflammatory nuclear factor kappa B (NF-κB) pathway [39,42]. Similarly, the sirtuin 1 (SERT1)/nuclear factor erythroid 2-related factor 2 (Nrf2) pathway was proposed to mediate the anti-inflammatory and anti-oxidative stress properties of incretin mimetics [39]. Decreased microglial activation achieved through these mechanisms decreased pro-inflammatory cytokine and ROS production, resulting in an attenuated immune response (Figure 1). This decreased inflammatory response helps reduce BBB permeability. Astrocyte aquaporin 4 (AQP4) polarization was also a proposed mechanism of action to stabilize BBB dysfunction and prevent cerebral edema [32]. However, studies found conflicting results on the effects of incretin mimetics on astrocytes. While some studies found that astrogliosis was decreased by incretin mimetic treatment [28,32], others found that decreases were not significant [17,24,33,34]. But some of these studies did acknowledge that their results could have differed due to experimental design [17,24,28,32,33]. Lastly, one study found that incretin mimetic treatment caused increased microglial activation and astrogliosis in certain parts of the brain after 6 weeks. The study proposed that microglial activity at later time points can be beneficial in promoting tissue repair and decreasing inflammation [34]. Ultimately, incretin mimetics’ anti-inflammatory and BBB stabilizing effects (Figure 1) were partly attributed to decreased cerebral edema, inflammation, neuronal degeneration, oxidative stress, and memory deficit outcomes.

## 4. Methodology

The articles discussed in this literature review are related to TBI, TBI complications, and neuroinflammation.

Articles were electronically retrieved through the NIH PubMed database.

Articles were screened by reviewing the title, abstract, and key words. Articles were included if they were relevant to the use of incretin mimetics for concussions, TBI, neuroinflammation or TBI sequelae (Table 3). Articles that were published from the period of 2012–2025 were considered for this review (Table 3).

Studies looking at the effects of incretin mimetics in models of stroke and ischemic brain injury were excluded (Table 3). Despite ischemic events being possible consequences of TBI brain injury, ischemia in these models was induced in a non-traumatic manner, making them not sufficiently related. Also, the studies looking at the effects of incretin mimetics in models of spinal cord injury were not included (Table 3). While both the spinal cord and brain are parts of the CNS, there was an absence of TBI in the spinal cord models, making them insufficiently related to the subject of the review. Studies in which brain injury was induced in a non-traumatic fashion, including non-TBI associated neuroinflammation, or studies using non-incretin interventions were excluded due to lack of relevance. Articles published before 2012 were excluded to keep reviewed material of current relevance (Table 3).

## 5. Conclusions and Future Directions

The results of these in vitro experiments and in vivo studies are quite promising for the future of incretin mimetic use in this field. However, there are gaps in knowledge present among the reviewed literature and areas that need to be further investigated. There are several effects that have not been concordant among all the studies reviewed. The exact effect of incretin mimetics on astrocytes is not clear, with studies finding differing results that could be attributed to differences in treatment timing, dose, or type of TBI. Our knowledge of how incretin mimetics affect astrocytes is vital, as they are a major contributor to the neuroinflammatory processes. Furthermore, although many studies concur that incretin mimetics exert their effects through pathways such as CREB, other pathways—such as MAPK—remain insufficiently explored, leaving questions about their role unanswered. Some studies used clinically translatable doses of incretin mimetics, but uncertainties still exist regarding what type of incretin mimetic and timing of treatment have maximal therapeutic benefits. Further investigation into these areas is instrumental for the future use of incretin mimetics for TBI and neuroinflammation.

These findings suggest incretin mimetics are promising candidates for TBI and concussion treatment. Incretin mimetics have the potential to address the injury mechanisms—excitotoxicity, oxidative stress, apoptosis—relevant to TBI. Additionally, they have a large safe use profile for T2DM and obesity [56,57]. However, incretin mimetic use for TBI and concussion treatment will require early-phase clinical studies that focus on specific windows of intervention (e.g., acute vs. subacute), carefully chosen populations (mild concussion vs. moderate/severe TBI), and validated outcome measures (cognitive composites, blood-based biomarkers such as GFAP/NfL, and advanced imaging readouts of BBB integrity or neuroinflammation). Future directions should also address potential limitations, including dosing strategies, treatment adherence in injured patients, and possible cardiovascular/metabolic side effects. Establishing feasibility in well-designed phase I/II trials would provide the necessary foundation for larger multicenter efficacy studies.

Finally, some of the articles reviewed also discussed the association between TBI and long-term neurodegenerative sequelae like Alzheimer’s disease. TBI is suggested to be causatively associated with neurodegenerative diseases through unresolved neuroinflammation and diffuse axonal injury [58]. Research for incretin mimetic use in neurodegenerative disorders is already more advanced than in TBI, with clinical trials for Alzheimer’s and Parkinson’s disease already underway [59,60,61,62]. The advancement of incretin mimetic use in neurodegenerative disease research to a clinical stage suggests a promising future for incretin mimetic use in TBI.

In conclusion, there is a necessity for more research on this subject, followed by clinical studies that will pave the way for the use of incretin mimetics in TBI and neuroinflammation. Based on experimental data discussed in the review, incretins have the potential to transform TBI and neuroinflammation treatment by reducing patient suffering and need for long term rehabilitation. Also, a personalized approach should be explored and patient populations that would benefit most from the treatment should be defined.

## Figures and Tables

**Figure 1 ijms-27-00045-f001:**
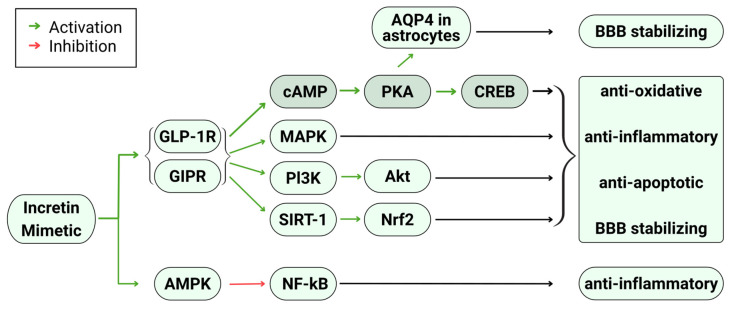
Incretin mimetics underlying mechanisms. Mechanistic pathways, discussed in the studies reviewed, that incretin mimetics used to achieve their neuroprotective and neurotrophic effects. The cAMP/PKA/CREB pathway is highlighted as it is the most discussed pathway among studies reviewed.

**Table 1 ijms-27-00045-t001:** Overview of studies reviewed.

Incretin Mimetic	Model	Outcomes	References
Experimental Protocols and Results			
Liraglutide. Commenced 30 min after injury and continued for 7 days.	Mild TBI murine model (mice)	Decreased spatial and visual memory impairment	Li et al. [23]
Liraglutide. Pre-treated 1 h before injury.	Human SH-SY5Y and SH-hGLP-1R#9 neuroblastoma cells and rat primary neuronal cultures	Induced cellular proliferation, recovered loss in cell viability due to excitotoxicity and oxidative stress, decreased caspase-3 activity, did not decrease apoptosis inducing factor (AIF) levels	Li et al. [23]
Liraglutide or Twincretin. Commenced 30 min after injury and continued for 7 days.	Mild TBI murine model (mice)	Decreased spatial and visual memory impairment, neuronal degeneration, and microglial activation but not astrogliosis, recovered mTBI induced p-PKA reduction	Bader et al. [24]
Exendin-4. Commenced immediately after trauma or pre-treated for 2 days. Continued for 7 days.	Mild TBI murine model (mice)	Decreased spatial and visual memory impairment, no significant improvement in passive avoidance	Rachmany et al. [25]
Exendin-4. Pre-treated shortly before injury or rapid treatment post insult.	Human SH-SY5Y neuroblastoma cells and rat primary neuronal cultures	Recovered loss in cell viability due to excitotoxicity and oxidative stress	Rachmany et al. [25]
Exendin-4. Pretreated for 2 days and continued for 7 days.	Mild TBI murine model (mice)	Decreased visual memory impairment, recovered altered gene expression	Tweedie et al. [26]
Twincretin. Commenced 30 min after injury and continued for 7 days.	Mild TBI murine model (mice)	Decreased spatial and visual memory impairment	Tamargo et al. [27]
Twincretin. Pre-treated 1 h before injury.	Human SH-SY5Y and SH-hGLP-1R#9 neuroblastoma cells and rat primary neuronal cultures	Increased cAMP levels and induced CREB pathway activation, recovered loss in cell viability due to excitotoxicity and oxidative stress	Tamargo et al. [27]
PT302. Commenced 1 h after injury and continued for 7 days.	Mild TBI murine model (rats and mice)	Decreased visual and spatial memory impairment, neuronal degeneration, and microglial activation but not astrogliosis	Bader et al. [17]
GIP. Pre-treated 2 days before injury. Continued for 14 days.	Mild TBI murine model (rats)	Decreased spatial and visual memory impairment, balance and fine motor impairment, astrogliosis, apoptosis, and axonal damage, did not affect sensorimotor impairment	Yu et al. [28]
Liraglutide. Commenced 10 min after injury. Continued 12, 24, and 36 h after injury.	Moderate-severe TBI murine model (rats)	Decreased sensorimotor impairment, cerebral edema, blood–brain barrier permeability, and cortical lesion size but not thalamic, did not reduce thalamic delayed neuronal cell death	Hakon et al. [29]
Exendin-4. Commenced 30 min after injury and continued for 7 days.	Moderate-severe TBI murine model (rats)	Decreased spatial memory impairment	Eakin et al. [30]
Exendin-4. Pre-treated 1 h before injury or rapid treatment post insult.	Human SH-SY5Y neuroblastoma cells and rat primary neuronal cultures	Recovered loss in cell viability due to excitotoxicity and oxidative stress, decreased caspase-3 activity	Eakin et al. [30]
GLP-1(7–36). Commenced immediately after injury and continued for 30 days.	Moderate-severe TBI murine model (rats)	Decreased neurological deficits, sensorimotor impairment, spatial memory impairment, cerebral edema, caspase-3 activity, and levels of hydrogen peroxide and reactive oxygen species, increased antioxidant factors, induced activation of the ERK5/CREB signaling pathways	Wang et al. [31]
Exendin-4. Commenced 1 h after injury and continued for 30 days.	Moderate-severe TBI murine model (mice)	Decreased spatial and visual memory impairment, blood–brain barrier dysfunction, astrogliosis, axonal injury, and caspase-3 activity, recovered glymphatic system dysfunction and aquaporin 4 polarization	Lv et al. [32]
Liraglutide. Commenced immediately after injury and continued for 3 days.	Moderate-severe TBI murine model (mice)	Decreased reactive oxygen and nitrogen species, inflammatory cytokines, lesion size, and apoptotic-induced caspase activity, did not affect astrogliosis, increased CREB activation and brain-derived neurotrophic factor	DellaValle et al. [33]
L-Carnitine and Exendin-4. Commenced immediately after injury. Continued for 14 days.	Moderate-severe TBI murine model (rats)	Decreased visual memory, sensorimotor and sensory function impairment, and oxidative stress, no decrease in microglial activation or astrogliosis, increased antioxidants, did not affect caspase-3 activity	Chen et al. [34]
Exendin-4. Commenced 2 h after injury or pre-treated for 2 days. Continued for 7 days.	Blast TBI murine model (mice)	Decreased neurodegeneration and visual memory impairment, recovered altered gene expression	Tweedie et al. [35]
Liraglutide. Commenced 2 h after injury or pre-treated for 2 days. Continued for 7 days.	Blast TBI chinchilla model	Increased hearing recovery, decreased caspase-3 activity	Jiang et al. [36]
Liraglutide. Commenced 2 h after injury or pre-treated for 2 days. Continued for 7 days.	Blast TBI chinchilla model	Increased hearing recovery	Jiang et al. [37]
Exendin-4. Commenced 2 h after injury or pre-treated for 2 days. Continued for 7 days.	Blast TBI murine model (mice)	Decreased visual and spatial memory impairment, recovered loss in synaptophysin immunoreactivity	Rachmany et al. [38]
Exendin-4. Pre-treated 2 h before injury	Mouse hippocampal HT22 cells	Recovered loss in cell viability and reduction in neurite length	Rachmany et al. [38]
Subarachnoid and Intracerebral Hemorrhage Models			
Semaglutide. Commenced immediately after injury. Continued for 2 days.	Subarachnoid hemorrhage murine model (mice)	Decreased cerebral edema, neuronal cell death, ferroptosis, oxidative stress, and pro-inflammatory cytokine expression	Chen et al. [39]
Liraglutide. Commenced 2 h after injury. Continued 12 h after injury.	Subarachnoid hemorrhage murine model (rats)	Decreased neurological function and sensorimotor deficit, cerebral edema, blood–brain barrier permeability, microglial activation, apoptosis, level of oxidative stress, pro-inflammatory cytokine expression, and caspase-3 activation	Tu et al. [40]
Exendin-4. Commenced 1 h after injury.	Subarachnoid hemorrhage murine model (rats)	Decreased neurological function, sensorimotor deficit, spatial memory deficit, and apoptosis	Xie et al. [41]
Liraglutide. Commenced 1 h after injury. Continued for 3 days.	Intracerebral hemorrhage murine model (mice)	Decreased cerebral edema, neurological function, sensorimotor deficit, and neutrophil infiltration and activation, increased cAMP levels	Hou et al. [42]

**Table 2 ijms-27-00045-t002:** Number of studies measuring common outcomes.

Outcome	Number of Articles	Model Type	
		In Vivo	In Vitro
Memory deficits	14	14	N/A
Excitotoxicity + oxidative stress	9	5	4
Inflammation + blood–brain barrier dysfunction	10	10	0
Sensorimotor impairment	7	7	N/A
Stimulation of anti-apoptotic and pro-survival pathways	10	8	2

**Table 3 ijms-27-00045-t003:** Inclusion and exclusion criteria and number of screened papers.

**Key Search Terms**	Incretin(s)	**Total Number of Screened Papers**	112
	incretin mimetic(s)		
	glucose-dependent insulinotropic peptide; GIP		
	glucose-dependent insulin-releasing hormone		
	gastric inhibitory peptide		
	glucagon-like peptide-1; GLP-1		
	glucagon receptor agonist(s)		
	glucagon		
	semaglutide		
	exenatide		
	liraglutide		
	exendin-4		
	dulaglutide		
	albiglutide		
	lixisenatide		
	tirzepatide		
	ozempic		
	traumatic brain injury; TBI		
	concussion		
	neuroinflammation		
**Screening Inclusion Criteria**	Incretins in concussions	**Number of Papers Included**	21
	Incretins in traumatic brain injury (TBI)		
	Incretins in TBI sequelae		
	Incretins in neuroinflammation associated with TBI		
	Incretins in TBI associated hemorrhage		
	Articles published from 2012–2025		
**Screening Exclusion Criteria**	Incretins in models of stroke	**Number of Papers Excluded**	91
	Incretins in models of ischemic brain injury		
	Non-traumatic induction of injury in model		
	Spinal cord injuries		
	Non-incretin intervention		
	Non-TBI associated neuroinflammation		
	Articles published before 2012		

## Data Availability

No new data were created or analyzed in this study. Data sharing is not applicable to this article.

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
