# Peer review of "Int. J. Mol. Sci.2026, 27(1), 45;https://doi.org/10.3390/ijms27010045"

_ijms, 2025, doi:10.3390/ijms27010045_

Round 1

Reviewer 1 Report

Comments and Suggestions for Authors

The review article "Incretin Mimetics as Potential Therapeutics for Concussion and Traumatic Brain Injury" addresses the highly relevant problem of developing effective therapies for traumatic brain injury (TBI). The authors present a compelling case for repurposing incretin mimetics - drugs with established safety profiles in diabetes and obesity - as promising neuroprotective agents. The authors provide a detailed and systematic description of TBI pathophysiology, including secondary injury mechanisms, neuroinflammation, oxidative stress, and blood-brain barrier dysfunction. The section on incretin mimetics' mechanisms of action is particularly thorough, clearly illustrating the key signaling pathways involved (cAMP/PKA/CREB, PI3K/Akt, MAPK) through which these drugs inhibit apoptosis, reduce excitotoxicity, and modulate inflammatory responses. The article deserves publication and will be valuable to both basic researchers and clinicians.

Some recommendations for improvement:

1. The methodology section is overly brief and lacks transparency. Although the manuscript is a narrative review, it is still advisable to describe the inclusion and exclusion criteria in more detail, and specify the number of studies ultimately included. This would enhance the credibility and reproducibility of the review.

2. A comparative analysis of incretin mimetics based on their key pharmacokinetic parameters could be added, and differences in the efficacy and safety of specific drugs could be discussed.

3. Since the ability to penetrate the blood-brain barrier is crucial for assessing the neurotropic potential of drugs, it would be useful to include existing data on the blood-brain barrier penetration of native incretins and their synthetic analogs in the review.

Reviewer 2 Report

Comments and Suggestions for Authors

The authors present a manuscript describing a literature review of Incretin Mimetics a potential therapeutics for Concussion and Traumatic Brain Injury (TBI). Some suggestions to improve:

Introdcution:

Page 1 lines 36-40: Citation needed

Page 1 line 41: Is this a correct reference for this statement

Results:

-In the current version, the results are given in the discussion section. There are only tables and a brief paragraph. There needs to be a description of the results in addition to the tables in this section.

Discussion:

The discussion should summarize the findings and describe their meaning and next steps.

Comments on the Quality of English Language

It would help for readability to have a English language review.

Round 2

Reviewer 2 Report

Comments and Suggestions for Authors

The authors appear to have comprehensively responded to reviewer feedback and updated the manuscript accordingly. It would be helpful to have an English language review. 

Comments on the Quality of English Language

It would help for readability to have a English language review.

Author Response

Comment: 

The authors appear to have comprehensively responded to reviewer feedback and updated the manuscript accordingly. It would be helpful to have an English language review.

Answer: We thank the reviewer for their comments and suggestions

Here we provide corrected version with improved English.

Also, we were notified from the Editorial Office that, if accepted, our manuscript will undergo professional language editing by their dedicated team as part of the standard publication process.